# Simultaneously Enhanced Thermal Conductivity and Dielectric Breakdown Strength in Sandwich AlN/Epoxy Composites

**DOI:** 10.3390/nano11081898

**Published:** 2021-07-24

**Authors:** Zhengdong Wang, Xiaozhuo Wang, Silong Wang, Jieyu He, Tong Zhang, Juan Wang, Guanglei Wu

**Affiliations:** 1School of Mechanical and Electrical Engineering, Xi’an University of Architecture and Technology, Xi’an 710055, China; wangzhengdong@xauat.edu.cn (Z.W.); wxz@xauat.edu.cn (X.W.); wsl@xauat.edu.cn (S.W.); hjy20010903@xauat.edu.cn (J.H.); zt@xauat.edu.cn (T.Z.); juanwang168@gmail.com (J.W.); 2Shaanxi Key Laboratory of Nano Materials and Technology, Xi’an University of Architecture and Technology, Xi’an 710055, China; 3State Key Laboratory of Bio-Fibers and Eco-Textiles, Institute of Materials for Energy and Environment, College of Materials Science and Engineering, Qingdao University, Qingdao 266071, China

**Keywords:** sandwich epoxy-based composites, aluminium nitride, dielectric properties, thermal conductivity

## Abstract

Polymer-based composites with high thermal conductivity and dielectric breakdown strength have gained increasing attention due to their significant application potential in both power electronic devices and power equipment. In this study, we successfully prepared novel sandwich AlN/epoxy composites with various layer thicknesses, showing simultaneously and remarkably enhanced dielectric breakdown strength and thermal conductivity. The most optimized sandwich composite, with an outer layer thickness of 120 μm and an inner layer thickness of 60 μm (abbreviated as 120-60) exhibits a high through-plane thermal conductivity of 0.754 W/(m·K) (4.1 times of epoxy) and has a dielectric breakdown strength of 69.7 kV/mm, 8.1% higher compared to that of epoxy. The sandwich composites also have higher in-plane thermal conductivity (1.88 W/(m·K) for 120-60) based on the novel parallel models. The sandwich composites with desirable thermal and electrical properties are very promising for application in power electronic devices and power equipment.

## 1. Introduction

Along with their rapid development within the electronic and electrical industry, high dielectric breakdown strength materials have attracted increasing attention due to their potential application in high-voltage equipment and power electronic devices with high power density (e.g., insulated gate bipolar transistor) [1,2,3,4]. Recently, polymer-based composites with high dielectric breakdown strength have been the focus of a large number of studies, due to their flexibility, low cost and processability [5,6,7,8,9]. Moreover, it is worth noting that these devices and equipment tend to be manufactured with high miniaturization and integration to improve performance, which leads to an increased threat from the high heat dissipation and electric field strength [10,11,12]. High thermal conductivity and dielectric breakdown strength polymer composites are, therefore, highly needed as dielectric materials for energy storage, power electronic devices, and power equipment to enhance heat conduction and the tolerance on electric field [13,14]. Epoxy resin is a material that is frequently used in electronic and electrical engineering applications, and has outstanding dielectric insulating properties, resistance to thermal and chemical stability, and a high glass transition temperature [15,16,17]. However, the low thermal conductivity of neat epoxy resin cannot meet the requirements of power electronics and equipment [18,19]. In order to enhance thermal conductivity, inorganic ceramic particles with high thermal conductivity can be incorporated into the polymer matrix [20,21,22,23]. In general, metals and carbon materials have high thermal conductivity [24,25,26]. However, the dilemma with employing these fillers is that the incremental gains in thermal conductivity are generally obtained at the cost of a substantial reduction in the dielectric breakdown strength of the polymer matrix, for which reason their applications in high-voltage equipment have been limited. It is, therefore, essential that novel materials are developed to achieve high thermal conductivity and acceptable dielectric breakdown strength.

Inorganic particles with high intrinsic thermal conductivity and a large band gap, such as Al_2_O_3_, BN and AlN, are usually used as fillers of polymer-based composites to enhance thermal conductivity and dielectric properties [27,28,29,30,31]. AlN is a good filler candidate due to its high thermal conductivity (320 W/(m·K)) and wide band gap (6.2 eV) [32]. To a large extent, the use of micro-sized fillers to improve the thermal conductivity of the epoxy matrix is appropriate due to the greater number of interfaces and phonon scattering of nanofillers [33,34]. However, it is more difficult for micro-sized fillers to form a compact and compatible interface due to their relatively low surface energy and specific surface area. The imperfect interfaces between filler and polymer lead to a strong decrease in the dielectric breakdown strength of polymer-based composites [8,35]. One method for resolving this issue is to graft organic groups onto the fillers [36,37,38], which can improve the interface interaction between the fillers and the polymer matrix, and simultaneously enhance dielectric breakdown strength and thermal conductivity. However, the thermal conductivity is enhanced by only a small amount, as the complex grafting process is difficult to control. In addition, interface defects lead to a significant reduction in dielectric breakdown strength. Therefore, achieving high thermal conductivity and dielectric breakdown strength in polymer composites remains a challenge. Another solution to improve the thermal conductivity and dielectric properties is to design a sandwich-structured polymer composite. In our previous work, we investigated the thermal conductivity and dielectric properties of epoxy-based composites with a sandwiched structure and an Al_2_O_3_ filler [6]. The results showed that the sandwiched composites have a higher dielectric breakdown strength and thermal conductivity due to electric field redistribution and the barrier effect, as a result of a changes in the initial breakdown conditions. However, the thermal conductivity of the composites remains relatively low due to the low thermal conductivity of the inner layer, which has a low content of Al_2_O_3_ filler.

In this study, AlN particles were used as fillers and layer thickness in the sandwiched composites was altered layer-by-layer via the solution casting and hot-press techniques. By controlling layer thickness, the thermal conductivity and dielectric breakdown strength could be regulated. In the sandwich-structured composites, the outer epoxy-based layers with micro-AlN showed an obvious increase in thermal conductivity. Meanwhile, the central layer with nano-AlN incorporated into epoxy matrix maintained a high dielectric breakdown strength. The effect of the outer and inner layer thickness on the dielectric properties and thermal conductivity of the laminated composites was also investigated, based on the electric field adjustment among the different layers and the barrier effect in the sandwich-structured composites. To our knowledge, sandwiched epoxy-based composites prepared by hot-press layer-by-layer have not yet been studied. The current study was expected to obtained a epoxy-based composite with significantly enhancing thermal conductivity and maintaining an acceptable dielectric breakdown strength through the design of sandwich-structured composites using the hot-press technique.

## 2. Experimental Methods

### 2.1. Materials

Nano-AlN with a diameter of 40 nm (defined as ANNP) and micro-AlN particles with an average size of 5 μm (defined as ANMP) were employed as inorganic fillers. Bisphenol-epoxy resin (DGEBA, E-828, HEXION) was purchased from Haoyun chemical Co. Ltd., Guangzhou, China. Methyl tetrahydrophthalic anhydride (MTHPA) and N, N-dimethylbenzylamine (BDMA) were supplied by Aladdin Reagent Co. Ltd., Shanghai, China, and were employed as a curing agent and accelerator, respectively.

### 2.2. Preparation of AlN/Epoxy Composites

For the fabrication of AlN/epoxy composites, a certain amount of AlN particles and epoxy monomers were added to a ball milling jar with zirconia balls for ultrahigh speed mixing and ball milling at 1000 rpm for 10 min and 2000 rpm for 20 min, via the planetary centrifugal mixer (Thinky, Laguna Hills, CA, USA). The zirconia balls were then removed from the slurry with the epoxy and fillers. The hardener MTHPA and accelerator BDMA were added into the slurry. The mixed solution was then dispersed by planetary centrifugal mixer at 1000 rpm for 5 min. The resultant suspension was moved to a vacuum oven for degassing at 60 °C for 30 min. The suspension was then poured into the middle of a tailored stainless steel spring sheet with a 10 cm square hole, with polyethylene terephthalate films used as a substrate on the top and bottom. A hot press of 20 MPa at 100 °C for 120 min was applied. Subsequently, the samples were moved to an oven and heated at 150 °C for 600 min for the post-curing process. The thickness of the composites was dependent on the thickness of the stainless steel spring sheet.

### 2.3. Preparation of the Sandwiched Epoxy-Based Composites

Laminated-structured epoxy-based composites were prepared layer-by-layer via the aforementioned hot-pressing method, as illustrated in Figure 1. The thickness of the films could be controlled by using stainless steel-spring sheets with different thicknesses. First, the ANMP/epoxy composite suspension was poured and formed into a film using the hot-press technique to create the first layer of the sandwich-structured composites. This first layer was subsequently dried at 100 ℃ for 30 min to form a stable shape. It is worth noting that the first layer should not be completely cured, so that it is able to be etched by the next solution layer. The epoxy-based composite solution with 1.0 vol.% ANNP was poured onto the first layer of dried ANMP/epoxy to create the middle layer. Similarly, the third layer of epoxy-based composite was fabricated by pouring ANMP/epoxy suspension onto the middle layer. The final sandwich composite was obtained after heating at 100 °C for 120 min by hot-press and post-curing at 150 °C for 600 min in the oven. The composites were peeled off to be free-standing, and were stored in a dry box for subsequent measurement. The sandwich composite with a 120 μm thick outer layer and 60 μm thick middle layer is defined as “120-60”, and all other samples named in the same way.

### 2.4. Characterizations

The phase structure of the materials was examined using X-ray diffraction (Bruker D2 PHASER (Karlsruhe, Germany). The surface morphology of the epoxy composites were studied using SEM (FEI QUANTA F250 Hillsboro, TX, USA. Laser flash apparatus (LFA467, Selb, Germany) was performed to test the thermal conductivity of samples. Dielectric properties of samples were measured by broadband dielectric analyzer (CONCEPT 80, Novocontrol Technology Company, Montabaur, Germany). Dielectric breakdown strength was studied using two sphere electrode systems. A layer of gold was sputtered on the surface of samples by an auto spraying equipment for the dielectric test. Differential scanning calorimetry (DSC, 200 F3) was carried out at the heating rate of 10 °C/min under a nitrogen atmosphere. 

## 3. Results and Discussions

### 3.1. Surface Morphology and Structure Analysis

In general, micro-sized fillers in polymer composites are beneficial for heat flux conduction due to lower amounts of photon scattering than that of nanofillers. Unfortunately, the dielectric breakdown strength of the epoxy composites with micro-fillers shows an obvious decrease in comparison with that of neat epoxy. In our previous work, multi-layer structured polymer composites were shown to concurrently enhance thermal conductivity and dielectric breakdown strength [6]. However, thermal conductivity was still low due to the intrinsic thermal conductivity of alumina and some defects between the micro-fillers and the epoxy matrix. In this work, AlN particles with high thermal conductivity were selected, and the layer-by-layer hot-pressing method was adopted to fabricate sandwich composites. The ANNP was used as a filler in the middle layer to ensure high dielectric breakdown strength, while the ANMP was mixed into the outer layers for thermal transport performance. The cross-section SEM image of the epoxy-based composite exhibits a clear multilayered structure, as shown in Figure 2a,d. The total thickness of the multi-layer composite is approximately 300 µm, with various layer thicknesses within the inner layer and outer layer, showing a distinguishing interface in the sandwich-structured composite. The perfect interfaces between adjoining layers are visible, and there are no obvious defects due to the compatible materials and etched surfaces. In other words, the surface of the only dried layer is easily etched by the solution of the next layer.

In the Figure 2b,c, the surface morphology of the epoxy composites indicated that the ANMP and ANNP were uniformly incorporated into the epoxy matrix, which was mainly attributed to the strong shear force applied during the high-speed ball mill mixing. Multilayered epoxy-based composites with different layer thicknesses were prepared, with 130-40 and 90-120, as shown in Figure 2e,f. All of these results demonstrate that the sandwich composites filled with ANNP and ANMP fillers can be fabricated by the layer-by-layer hot-press technique. It is also worth noting that ANMP particles with an average size of 5 μm could also form compatible and compact interfaces with the epoxy matrix by using the ball milling treatment.

### 3.2. Thermal Conductivity and Dielectric Breakdown Strength

The through-plane thermal conductivity (TPTC) of the sandwich composites with various layer thicknesses at different test temperatures is shown in Figure 3a. The TPTC of the sandwich composites gradually increases with the increment of thickness of the outer layer. For instance, the TPTC of 120-60 reaches up to 0.754 W/((m·K)), which is 4.1 times higher compared to that of the epoxy. A higher value of 0.904 W/((m·K)) is obtained when the thickness of the outer layer is 130 μm. Dielectric breakdown strength is of great importance to power electronic devices and power equipment in practical applications. The dielectric breakdown strength values were analyzed by Weibull distribution. In the sandwich composites, 1.0 vol.% loading of ANNP was embedded into the inner layer, as the composite exhibits the highest dielectric breakdown strength in Appendix A. The 35.6 vol.% of inorganic fillers in the outer layers of epoxy composites were chosen due to the simultaneous consideration of the thermal conductivity, dielectric constant and dielectric breakdown strength. Based on our previous study [6], the epoxy composites with 35.6 vol.% (corresponding to 60 wt.%) in outer layers showed enhancements in both thermal conductivity and dielectric breakdown strength compared to other filler loadings in the outer layers. When the filler loading of the epoxy composites is less than 35.6 vol.%, thermal conductivity is relatively low. However, the epoxy composites with more than 35.6 vol.% fillers become increasingly viscous, and the dielectric breakdown strength becomes significantly degraded. The composites with less than 35.6 vol.% fillers showed the desirable networks, viscosity, dielectric constant and electric field distribution, giving rise to the optimized parameters of thermal conductivity and dielectric breakdown strength. Figure 3b shows the weibull distribution of characteristic dielectric breakdown strength values. The dielectric breakdown strength of the sandwich composites first increases and then decreases with each increment of thickness in the outer layers. The 120-80 composite has an optimal dielectric breakdown strength value of 69.1 kV/mm, an increase of 8.1% in comparison with pure epoxy. For ease of comparison, the comprehensive contrast between thermal conductivity and dielectric breakdown strength is shown in Figure 3c. The dielectric breakdown strength of the 1.0 vol.% ANNP/epoxy composite is 66.5 kV/mm. However, its thermal conductivity is only 0.20 W/(m·K). It is evident that the dielectric breakdown strength and TPTC of the sandwich composites are higher and can be adjusted by layer thickness. For instance, the TPTC and dielectric breakdown strength of 120-60 are 0.754 W/(m·K) and 69.7 kV/mm, respectively. More notably, the TPTC of 130-60 reaches 0.904 W/(m·K), and its dielectric breakdown strength is still close to that of pure epoxy.

Electric field strength has a close correlation with the dielectric constant in terms of double-layer dielectric capacitors based on the series model, i.e., *E*_m_·*ε*_m_ = *E*_n_·*ε*_n_, and the principle also works in sandwich dielectric materials. Therefore, the electric field distribution of sandwich composites could be regulated by altering the dielectric constant of layer thicknesses. Based on the correlation between electric field strength, voltage, and thickness, the electric field strength of the sandwich composites is given by Equations (1) and (2),
(1)$$E_{m}=V/\left(2d_{m}+d_{n}\varepsilon_{m}/\varepsilon_{n}\right)$$
(2)$$E_{n}=V/\left(d_{n}+2d_{m}\varepsilon_{n}/\varepsilon_{m}\right)$$
where *V* is the applied voltage; *d*_n_ and *d*_m_ is the thickness, *ε*_n_ and *ε*_m_ represents permittivity, *E*_n_ and *E*_m_ represents electric field strength, the subscript m and n represent outer layer and inner layer, respectively.

Based on the equations mentioned above, the ANNP/epoxy in inner layer would suffer a higher electric field strength due to the lower permittivity, when the layer thicknesses are the same. In other words, the outer layers suffered a lower electric field strength, leading to an incomplete breakdown of the sandwich composites. Moreover, the charges that gathered in the interfaces, as illustrated in Figure 3d, could be released in the incomplete breakdown channels, which is favorable for reducing dielectric stress and avoiding a complete breakdown. Another reason for this is that the dielectric breakdown strength values of samples increased with decreasing thicknesses, as shown in Appendix A. That is also why 120-60 showed the highest dielectric breakdown strength instead of 90-120. It means that the dielectric breakdown strength should be determined by layer thickness and the redistribution of electric field strength. When the total thicknesses of the sandwich composite were fixed, the variation in thickness of the inner layer and outer layer could alter the dielectric constant in each layer, determining the distribution of the electric field. Therefore, dielectric breakdown strength could be enhanced at the optimum layer thickness. In addition, the thermal transport process in the in-plane direction of the sandwich composite was revealed in Figure 3d. The thermal transport performance of sandwich composites should be higher because the heat flow can be more effectively conducted in the ANMP/epoxy outer layers.

In order to further analyze the thermal conductivity of laminated composites, a modified series model and a modified parallel model are presented.

Modified series model:(3)$$\frac{1}{K_{c}}=\frac{\Phi_{1}}{K_{1}}+\frac{\Phi_{2}}{K_{2}}+\frac{\Phi_{3}}{K_{3}}$$

Modified parallel model:(4)$$K_{c}=\Phi_{1}K_{1}+\Phi_{2}K_{2}+\Phi_{3}K_{3}$$
(5)$$\Phi_{1}+\Phi_{2}+\Phi_{3}=1$$
where *K*_c_ stands for thermal conductivity of sandwich composites, *K*_1_, *K*_2_ and *K*_3_ represent thermal conductivity of upper, inner and under layer, respectively. *Φ*_1_, *Φ*_2_ and *Φ*_3_ are the filler volume fraction of upper, inner and under layer, respectively.

In Figure 4, the experimental and predicted values of the thermal conductivity of laminated composites were compared. Thermal conductivity values of the sandwich composites were calculated by combining Equations (3) and (5). *K*_1_, *K*_2_ and *K*_3_ were obtained from the experimental data of the epoxy composites, summarized in Appendix A. *Φ*_1_, *Φ*_2_ and *Φ*_3_ were proportional to the thickness of each layer in the sandwich composites, and are also summarized in Appendix A. In this study, the thicknesses of the outer layers in the sandwich composites are equal; thus, the *Φ*_1_ and *Φ*_3_ are also equal. The SEM images have indicated that the sandwich composite with a perfect interface between adjacent layers could be considered as a whole, where contact thermal resistance could be neglected. Therefore, the TPTC of the sandwich composite is equivalent to that of the three layers in series. Similarly, the in-plane thermal conductivity (IPTC) of the composite is equivalent to that of the three layers in parallel. It is worth noting that the predicted thermal conductivity along the in-plane direction is much higher than that along the through-plane direction. For example, the predicted IPTC of 120-60 is 1.88 W/((m·K)), 2.5 times higher than the TPTC. This could be attributed to heat flow having to cross through the inner layer with low thermal conductivity for TPTC. However, for IPTC, heat flux could transfer directly along the outer layers, significantly enhancing thermal conduction efficiency.

### 3.3. Dielectric Constant and Dielectric Loss

The dielectric constant and dielectric loss are also important parameters for practical application in power electronic devices and power equipment. The dielectric behavior of sandwich composites with different layer thicknesses was studied, as showed in Figure 5. As can be seen from Figure 5a, the permittivity of sandwich structured composites gradually increases with each increment of layer thickness in the outer layers. For instance, the dielectric constant varies from 5.1 for 90-120 to 6.0 for 130-40 at 10^−1^ Hz. Based on previous reports, charges may be stored at the interfaces between the hetero layers within the sandwich composite [39,40,41]. Figure 5b shows the dependence of the dielectric constant and dielectric loss on layer thickness in all sandwich composites. It can be seen that the dielectric loss in all samples remains relatively low, and has no dramatic variation between the various layer thicknesses. This could be attributed to two reasons. First is that dielectric loss in the outer layer ANMP/epoxy composites depend on the interfaces and compatibility between the fillers and the epoxy matrix. The composites prepared with the ball-milling method show relatively lower dielectric loss, due to more compact interfaces and better compatibility. The more compact interfaces and better compatibility suppresses the mobility of charge carriers and polymer molecular swing, reducing interface relaxation loss. Secondly, the inner layer of the sandwich-structured composite can hinder the effective conduction of charges between two electrodes, giving rise to decrease of electrical conduction in the composite. This is due to the ANNP fillers incorporated into the inner layer of the sandwich composites having a small effect, resulting in lower electrical conductivity and a stronger interface interaction. In this case, the interface loss and electrical conduction loss were significantly reduced, and thus the dielectric loss was visibly improved. Moreover, dielectric losses for sandwich composites have a slight dependence on the thickness of inner and outer layers. Dielectric losses slightly increase with increasing outer layer thickness, implying that charges can be effectively hindered even though the inner layer is thin, which is advantageous in the design of materials with low dielectric loss.

## 4. Conclusions

In summary, we reported on the preparation of laminated composites by incorporating ANNP in the inner layers and ANMP in the outer layers using layer-by-layer solution pouring and the hot-pressing technique. The sandwich composites show many improved properties. For instance, the dielectric breakdown strength of 120-80 is up to 65.3 kV/mm, increased by 26% compared to that of traditional composites. Meanwhile, the laminated composites still have high TPTC and extremely low dielectric loss. More interestingly, the thermal conductivity of laminated composites could be predicted by a series model and parallel model. The predicted IPTC of 120-80 is up to 0.902 W/((m·K)), higher than TPTC. The sandwich epoxy composites are very promising for application in power electronic devices and power equipment.

## Figures and Tables

**Figure 1 nanomaterials-11-01898-f001:**
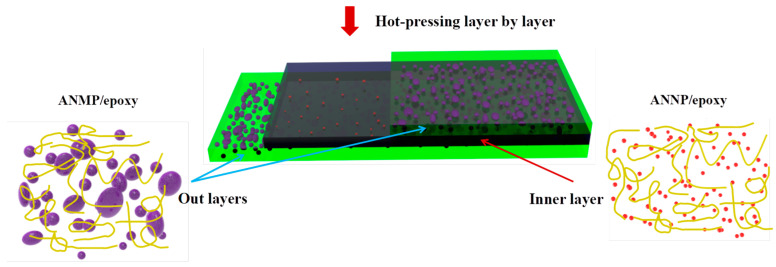
Schematic of the preparation process of sandwich-structural epoxy-based composites.

**Figure 2 nanomaterials-11-01898-f002:**
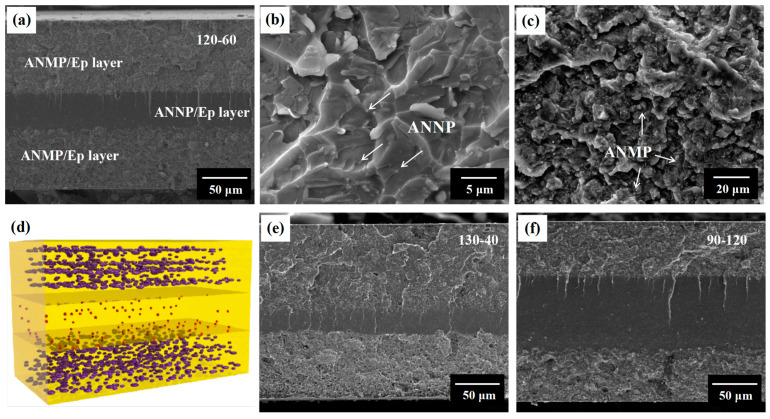
Cross-section SEM images of (**a**) 120-60, (**e**) 130-40 and (**f**) 90-120; Corresponding enlarged SEM images of (**b**) the inner layer with 1.0 vol.% ANNP in 120-60 and (**c**) the outer layer with 35.6 vol.% ANMP in 120-60; (**d**) schematic of the sandwiched composite.

**Figure 3 nanomaterials-11-01898-f003:**
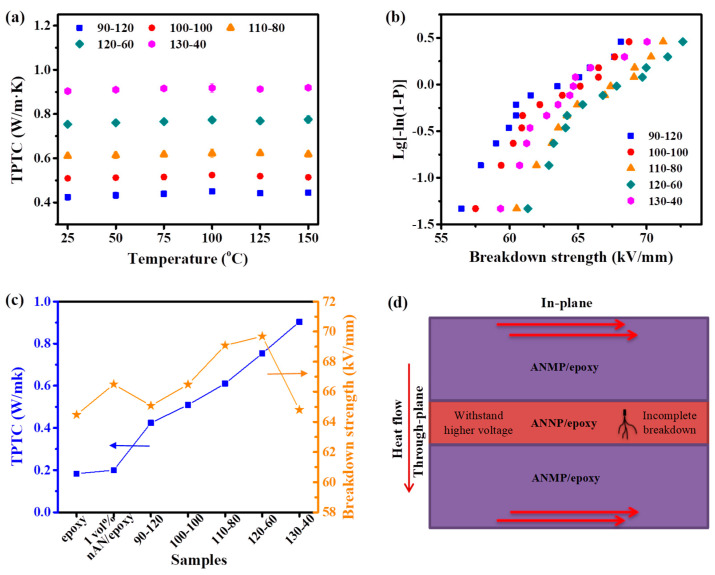
(**a**) TPTC of sandwich composites at different temperatures, (**b**) Weibull distribution for breakdown strength of sandwich composites, (**c**) comprehensive contrast of thermal conductivity and dielectric breakdown strength, (**d**) schematic of thermal transport and breakdown in sandwich composite.

**Figure 4 nanomaterials-11-01898-f004:**
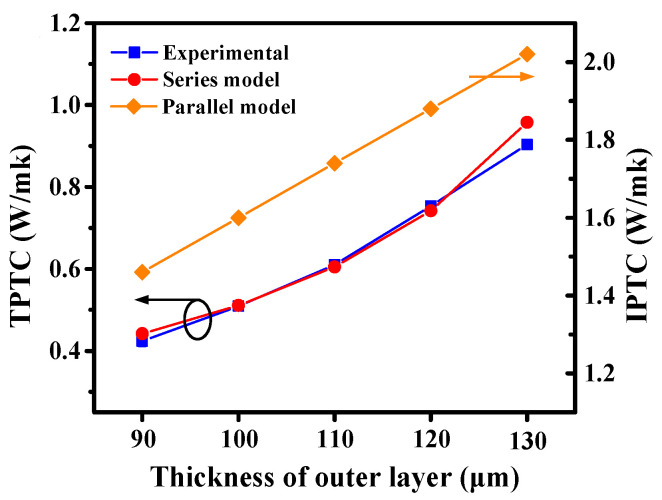
The experimental and theoretical thermal conductivity in a sandwich composite.

**Figure 5 nanomaterials-11-01898-f005:**
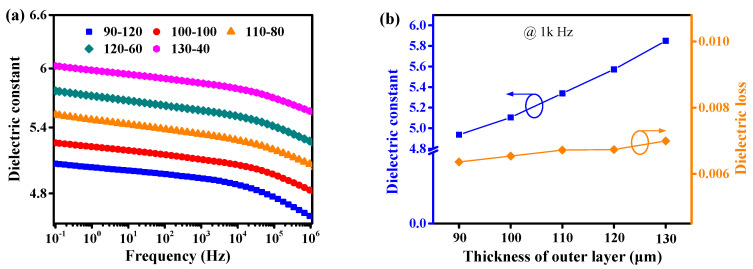
(**a**) dependence of frequency on dielectric constant of sandwich composites, (**b**) dependence of thickness in outer layers on dielectric loss of sandwich composites.

## Data Availability

Data is contained within the article.

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
