# Peer review of "Simultaneously Enhanced Thermal Conductivity and Dielectric Breakdown Strength in Sandwich AlN/Epoxy Composites"

_nanomaterials, 2021, doi:10.3390/nano11081898_

Round 1

Reviewer 1 Report

The paper introduces a well-organized research work on thermal conductivity and dielectric breakdown strength of sandwich AlN/epoxy nanocomposites. 
The title of the manuscript is informative and relevant. Abstract match the rest of the article. The introduction section is well organized and presents motivation and proper background study. It is clear what is already known about the AlN/epoxy nanocomposites. The authors emphasize the limitation of the previously prepared compounds. The research work is very well organized and structured. During the preparation of the manuscript, the authors followed the widely used IMRaD structure, it is easy to read, and the main research flow is clear. 
The materials and methods are introduced in enough detail, enabling reproducibility of the results. The study methods are valid and reliable.
Figures are relevant and clearly presented. 
The Conclusion is based on the result of the study and answers the research aim.

One remark: The text contains some typo errors please read through the manuscript carefully, and correct them.

Author Response

Response to Reviewer's Comments:

Reviewer 1: The paper introduces a well-organized research work on thermal conductivity and dielectric breakdown strength of sandwich AlN/epoxy nanocomposites. The title of the manuscript is informative and relevant. Abstract match the rest of the article. The introduction section is well organized and presents motivation and proper background study. It is clear what is already known about the AlN/epoxy nanocomposites. The authors emphasize the limitation of the previously prepared compounds. The research work is very well organized and structured. During the preparation of the manuscript, the authors followed the widely used IMRaD structure, it is easy to read, and the main research flow is clear. The materials and methods are introduced in enough detail, enabling reproducibility of the results. The study methods are valid and reliable. Figures are relevant and clearly presented. The Conclusion is based on the result of the study and answers the research aim.

One remark: The text contains some typo errors please read through the manuscript carefully, and correct them.

Answer: Thank you for your valuable suggestions/ comments very much. As suggested by the referee, we have already checked and revised our manuscript carefully. For instance, the “d” in the Fig. 2f has been revised to “f”. And thickness of the scale bar in Fig. 2e has been revised to be same as those of other figures. The caption “Fig. 5 Frequency dependence of (a) dielectric constant and (b) dielectric loss of sandwich composites.” is wrong. The caption has been revised to “Fig. 5 (a) dependence of frequency on dielectric constant of sandwich composites, (b) dependence of thickness in outer layers on dielectric loss of sandwich composites.”. And the corresponding caption of Fig. 5 has been marked with red color and underline. In 2.3 Section, the words “hot pressing” has been revised to “hot-pressing”, and “a illustrated” has been revised to “an illustrated”. In “Introduction” Section, the “a sandwich-structure polymer composites” and “the composites has a higher” were corrected to “a sandwich-structure polymer composite” and “the composites have a higher”. Moreover, “inner layer and Al2O3 filler” was revised to “inner layer with low content of Al2O3 filler”. And there are some other typo errors and grammar problems, which they were not listed here, but we have revised directly on the manuscript with red color and underline.

Reviewer 2 Report

Authors report the thermal and dielectric properties of AlN/epoxy composites. The simultaneous enhancements are a great interest of power electronic applications. 
1. However, the current manuscript does not describe much information on the physical pictures of the composite materials. For instance, they compared the properties with different thicknesses of the inner and outer layers of the multilayered system, but there is a lack of discussions on why authors chose such dimensions of thickness combinations, fractions of inorganic filers, and epoxy, etc. 
2. Insufficient references were cited to describe the current state-of-the-art properties of composites. 
3. Furthermore, it is expected to have non-negligible thermal interface resistances between AlN and epoxy, as well as between the neighboring layers, but the introduced thermal model does not include the interfacial effects. No measurement protocols have been introduced. 
4. Lastly, the entire manuscript consists of inconsistent font sizes, styles, and unit typos in figures.

Author Response

Response to Reviewer's Comments:

Reviewer 2: Authors report the thermal and dielectric properties of AlN/epoxy composites. The simultaneous enhancements are a great interest of power electronic applications.

  1. However, the current manuscript does not describe much information on the physical pictures of the composite materials. For instance, they compared the properties with different thicknesses of the inner and outer layers of the multilayered system, but there is a lack of discussions on why authors chose such dimensions of thickness combinations, fractions of inorganic filers, and epoxy, etc.

Answer: Thank you for your valuable suggestions/ comments very much. The discussions on dimensions of thickness combinations, fractions of inorganic filers, and epoxy were added.

In our work, the total thicknesses of 300 microns was selected because of the measuring requirements of thermal conductivity via laser flash apparatus (FLA467). When the thicknesses of samples were less than 300 microns, their through-plane thermal conductivity is lower than their intrinsic value. For instance, thermal conductivity of pure epoxy resin film with the thickness of 100 microns is only 0.162 W/(m·K), which is much lower than 0.185 W/(m·K) of epoxy resin. And the epoxy resin with he thicknesses of 300 microns has a close thermal conductivity of 0.183 W/(m·K) compared to that of epoxy resin. Therefore, the total thicknesses of 300 microns were designed in this work.

The 1.0 % volume fractions of inorganic fillers in the inner layer of epoxy composites were chosen because the epoxy composite with 1.0 vol.% of AlN showed the highest dielectric breakdown strength in Table S1. We added the description on the dielectric breakdown strength in 3.2 Section (3.2 Thermal conductivity and dielectric breakdown strength) and the words have been marked with red colour and underline. The 35.6 vol.% of inorganic fillers in the outer layers of epoxy composites were chosen because the thermal conductivity, dielectric constant and dielectric breakdown strength were simultaneously considered. Based on our previous study[6], the epoxy composites with 35.6 vol.% (corresponding to 60 wt.%) in outer layers showed simultaneous enhancements on thermal conductivity and dielectric breakdown strength compared to other filler loading in outer layers. When the epoxy composites with less than 35.6 vol.% fillers, thermal conductivity is relatively low. However, the epoxy composites with more than 60 wt.% fillers, the composites would begin to be more sticky and the dielectric breakdown strength would become significantly degraded. When the composites with less than 35.6 vol.% fillers showed the desirable networks, viscosity, dielectric constant and electric field distribution, giving rise to the optimized parameters of thermal conductivity and dielectric breakdown strength. The corresponding descriptions have been added into the manuscript and marked with red colour and underline.

The reason that epoxy resin have been explained briefly in the Introduction Section, and the relevant descriptions have been marked with red colour and underline.

  1. Insufficient references were cited to describe the current state-of-the-art properties of composites.

Answer: As suggested by the referee, the extra references were cited to show a more comprehensive understanding about the current state-of-the-art properties of composites (please see them in below). And the corresponding references have been marked with red colour and underline in References Section. The most of these references added were “Review”, which they are more comprehensive and important to reflect the he current state-of-the-art properties of composites.

5. Daniel,T. The search for enhanced dielectric strength of polymer-based dielectrics: A focused review on polymer nanocomposites[J]. J Appl. Polym. Sci. 2020 137(33), 49379.

13. Lokanathan, M.; Acharya, P.V.; Ouroua, A.; Strank, S.M.; Hebner, R.E.; Bahadur, V. Review of Nanocomposite Dielectric Materials With High Thermal Conductivity[J]. P IEEE 2021, 109, 1364-1397.

14. Zhang, L.; Deng, H.; Fu, Q.; Recent progress on thermal conductive and electrical insulating polymer composites[J]. Compos. Commun. 2018, 8, 74-82.

15. Bansal, R.K.; Sahoo, C. Thermal and dielectric properties of epoxy resins[J]. Angew. Makromol. Chem. 1979, 79, 125-132.

18. Wan, Y.-J.; Li, G.; Yao Y.-M.; Zeng, X.-L.; Zhu, P.-L.; Sun, R. Recent advances in polymer-based electronic packaging materials[J]. Compos. Commun. 2020, 19, 154-167.

20. Meziani, M. J.; Song, W.-L.; Wang, P.; Lu, F. S.; Hou, Z. L.; Anderson, A.; Maimaiti, H.; Sun, Y.-P. Boron nitride nanomaterials for thermal management applications[J]. ChemPhysChem 2015, 16, 1339-1346.

22. Huang, X.Y.; Jiang, P.K.; Tanaka, T. A review of dielectric polymer composites with high thermal conductivity[J]. IEEE Electr. Insul. M. 2011, 27, 8-16.

23. Guo, Y.Q.; Ruan, K.Q.; Shi, X.T. Yang, X.T.; Gu, J.W. Factors affecting thermal conductivities of the polymers and polymer composites: A review[J]. Compos. Sci. Techno. 2020, 193, 108134.

  1. Furthermore, it is expected to have non-negligible thermal interface resistances between AlN and epoxy, as well as between the neighboring layers, but the introduced thermal model does not include the interfacial effects. No measurement protocols have been introduced.

Answer: Thank you for your valuable suggestions/ comments very much. The thermal interface resistances between AlN and epoxy were not considered in our thermal conductivity model because the thermal conductivity of each layer in the epoxy composite has been independently measured. And the thermal conductivity in the model was calculated by using the measured thermal conductivity of each layer instead of the predicted one. Therefore, the thermal interface resistances between AlN and epoxy would not be considered in our model.

The thermal interface resistances between the neighboring layers should have been considered. However, the thermal interface resistances will be very difficult to be measured to obtain a accurate value because the current equipment or measurement protocols is limited and the thermal interface resistances is located in the interior of multilayered system. Fortunately, the multilayered composites can be seen as a whole one because the well-welding and prefect interfaces between the outer layers and inner layer have been formed. Therefore, the thermal interface resistances between the neighboring layers were considered to be very small. Based on the mentioned-above reasons, the thermal interface resistances between the neighboring layers were not considered. And the model can give a relatively rough prediction on thermal conductivity.

  1. Lastly, the entire manuscript consists of inconsistent font sizes, styles, and unit typos in figures.

Answer: Thank you for your valuable suggestions/ comments very much. As suggested by the referee, we have already checked and revised our manuscript carefully. For instance, the “d” in the Fig. 2f has been revised to “f”. And thickness of the scale bar in Fig. 2e has been revised to be same as those of other figures. The inconsistent font sizes in the caption of Fig. 5 have been revised to the uniform font size for the captions of figure and references. The caption “Fig. 5 Frequency dependence of (a) dielectric constant and (b) dielectric loss of sandwich composites.” was described wrongly. The caption has been revised to “Fig. 5 (a) dependence of frequency on dielectric constant of sandwich composites, (b) dependence of thickness in outer layers on dielectric loss of sandwich composites.”. And the corresponding caption of Fig. 5 has been marked with red color and underline. In 2.3 Section, the words “hot pressing” has been revised to “hot-pressing”, and “a illustrated” has been revised to “an illustrated”. In “Introduction” Section, the “a sandwich-structure polymer composites” and “the composites has a higher” were corrected to “a sandwich-structure polymer composite” and “the composites have a higher”. Moreover, “inner layer and Al2O3 filler” was revised to “inner layer with low content of Al2O3 filler”. And there are some other inconsistent font sizes, styles, and unit typos in figures, which they were not listed here, but we have revised directly on the manuscript with red color and underline.

Round 2

Reviewer 2 Report

The manuscript has been properly revised.